covid19.Explorer: a web application and R package to explore United States COVID-19 data

http://orcid.org/0000-0003-0767-4713 Revell Liam J. 1 2 liam.revell@umb.edu
1 Department of Biology, University of Massachusetts at Boston , Boston, MA , USA
2 Facultad de Ciencias, Universidad Católica de la Santísima Concepción , Concepción , Chile
Zuniga-Gonzalez Carlos
Electronic publication date: 2021 Aug 20
Publication date: 2021
Volume: 9
Electronic Location ID: e11489
Received 2021 Feb 16; Accepted 2021 Apr 27
Copyright: © 2021 Revell
Copyright year: 2021
Copyright holder: Revell
License: This is an open access article distributed under the terms of the Creative Commons Attribution License, which permits unrestricted use, distribution, reproduction and adaptation in any medium and for any purpose provided that it is properly attributed. For attribution, the original author(s), title, publication source (PeerJ) and either DOI or URL of the article must be cited.
License URL: https://creativecommons.org/licenses/by/4.0/

Keywords: Epidemiology, Infectious disease, SARS-CoV-2, Software

Funding: National Science Foundation DBI-1759940 FONDECYT, Chile 1201869 This research was funded by grants from the National Science Foundation (DBI-1759940) and FONDECYT, Chile (1201869). The funders had no role in study design, data collection and analysis, decision to publish, or preparation of the manuscript.

==============================
Appearing at the end of 2019, a novel virus (later identified as SARS-CoV-2) was characterized in the city of Wuhan in Hubei Province, China. As of the time of writing, the disease caused by this virus (known as COVID-19) has already resulted in over three million deaths worldwide. SARS-CoV-2 infections and deaths, however, have been highly unevenly distributed among age groups, sexes, countries, and jurisdictions over the course of the pandemic. Herein, I present a tool (the covid19.Explorer R package and web application) that has been designed to explore and analyze publicly available United States COVID-19 infection and death data from the 2020/21 U.S. SARS-CoV-2 pandemic. The analyses and visualizations that this R package and web application facilitate can help users better comprehend the geographic progress of the pandemic, the effectiveness of non-pharmaceutical interventions (such as lockdowns and other measures, which have varied widely among U.S. states), and the relative risks posed by COVID-19 to different age groups within the U.S. population. The end result is an interactive tool that will help its users develop an improved understanding of the temporal and geographic dynamics of the SARS-CoV-2 pandemic, accessible to lay people and scientists alike.

Introduction

In 2019, a novel infectious disease was identified in Wuhan, a city of approximately 11 million residents located in the Hubei Province of central China. This infectious disease, called Coronavirus disease 2019, or COVID-19 (Velavan & Meyer, 2020), is now known to be caused by the previously unidentified Severe Acute Respiratory Syndrome Coronavirus 2 or SARS-CoV-2 (Wong et al., 2020). Following the Wuhan outbreak, cases of SARS-CoV-2 infection and COVID-19 death were subsequently identified in Europe, the United States, and (by the time of writing) at least 192 countries worldwide. Counting from the beginning of this global pandemic, there have been nearly 3.1 million confirmed COVID-19 deaths, more than 580,000 of which have occurred in the United States alone (Centers for Disease Control and Prevention (CDC), 2021).

R (R Development Core Team, 2020) is a powerful scientific computing environment and programming language that is used by statisticians, data scientists, academic researchers, and students worldwide. I have built a multifunctional R package (covid19.Explorer) and corresponding web application (https://covid19-explorer.org). The purpose of both is to aid scientists and members of the public alike to better understand the 2020/21 SARS-CoV-2 pandemic in the United States. Although my focus is on U.S. COVID-19 data, readers from other countries might also be interested in the project—for instance, because the seasonal dynamics of infection or the age distribution of mortality has been broadly similar among different affected areas of the globe (e.g., Fantin, Brenes-Camacho & Barboza-Solıs, 2021; Hradsky & Komarek, 2021; Liu et al., 2021a; but see Hoffmann & Wolf, 2020; Gardiner, Oben & Sutcliffe, 2021).

This R package and website are not designed to be a substitute or replacement for the many other excellent software products and web tools that have been developed over the past year (e.g., Brown et al., 2020; Johns Hopkins University, 2020; Reiner, Barber & Murray, 2020; Gu, 2020). It nonetheless contains a number of different analytical approaches and methods that distinguish it from other software and web resources.

For example, the covid19.Explorer R package is the only software that I know of that allows the user to specify a custom model of the infection fatality ratio (IFR), the fraction of all SARS-CoV-2 infected individuals that ultimately die of COVID-19 in a given population (Blackburn et al., 2021), through time and then uses this model to reconstruct daily SARS-CoV-2 infections. Although this strategy has been employed by other modelers to estimate daily SARS-CoV-2 infections throughout the pandemic—most notably, perhaps, by Gu (2020); though other modeling groups also use confirmed daily COVID-19 deaths as an important lagging indicator of new infections (e.g., Reiner, Barber & Murray, 2020)—mine is, so far as I am aware, the only software that puts this model of IFR entirely under user control.

Likewise, the covid19.Explorer R package and website includes visualization methods not available in other software or web resources. For instance, the covid19.Explorer can create a plot of U.S. state-wise daily estimated infections in aggregate that is unlike any graphical representation of United States SARS-CoV-2 infection data that I have seen in other software, webpages, or media sources. Similarly, the package includes an ‘iceberg graph’ showing daily observed SARS-CoV-2 infections above the waterline, and estimated unobserved infections below it. I have likewise never encountered a precisely identical visual representation of the U.S. COVID-19 pandemic data in other electronic resources or software.

Lastly, it’s perhaps important to mention one thing that the covid19.Explorer R package most adamantly does not do, and that is make predictions about the future. There are numerous different individual scientists and research teams that have dedicated enormous effort and resources to predicting the epidemic dynamics of SARS-CoV-2 in the United States and globally (e.g., Reiner, Barber & Murray, 2020; Gu, 2020) with widely varying success (e.g., Chin et al., 2020; Ioannidis, Cripps & Tanner, 2020; James et al., 2021). The covid19.Explorer R package and site have the more modest goal of helping users develop a better understanding of what has happened over the course of the United States SARS-CoV-2 pandemic from its beginnings to the present day.

Methods

Preamble

covid19.Explorer is a library of functions and data that can be loaded and run using the R scientific computing software (R Development Core Team, 2020). The covid19.Explorer package is open source and freely available from its GitHub page (https://github.com/liamrevell/covid19.Explorer/). The covid19.Explorer package in turn depends on the CRAN R packages maps (Becker et al., 2018), phytools (Revell, 2012), randomcoloR (Ammar, 2019), and RColorBrewer (Neuwirth, 2014).

Though the covid19.Explorer R package can be downloaded, installed, and run from R on its own, it has primarily been designed to be utilized via a web portal: https://covid19-explorer.org. This web portal was built in the integrated development environment Rstudio (RStudio Team, 2020), using the web application development system shiny (Chang et al., 2021). In addition to those R libraries already mentioned, the web application also uses the package shinyWidgets (Perrier, Meyer & Granjon, 2021).

The data used by the various applications of the covid19.Explorer are all publicly available and were obtained (unless otherwise indicated) from the United States Centers for Disease Control and Prevention National Center for Health Statistics (https://www.cdc.gov/nchs/; henceforward, the CDC) or the United States Census Bureau (https://www.census.gov/). In particular, these data consist of: provisional U.S. COVID-19 death counts by sex, age, and week from the CDC; United States confirmed COVID-19 cases and deaths by state through time from the CDC; weekly counts of deaths by jurisdiction and age group from the CDC, 2015–present; weekly counts of deaths by state and select causes (including COVID-19) from 2014–2018, 2019–2020, and 2020–2021 from the CDC; estimated population sizes by U.S. state and by age, from 2010–2019 from the Census Bureau; and, finally, the geographic center of each U.S. state (to be used for mapping visualizations).

Types of functions in covid19.Explorer

The covid19.Explorer R package (and corresponding web application) consists of two main types of functions.

The first of these (exemplified by the shiny web application webpage tabs denominated U.S. COVID-19 infections, Iceberg plot, State comparison, Plausible range, and Infection estimator) consists of functions that are designed to estimate the true number of COVID-19 infections over the course of the pandemic. Since there are a variety of reasons that the true number of infections (rather than simply the number of confirmed cases) is of interest, these various aforementioned applications of the covid19.Explorer package are all designed to help users apply a model (of their own design, see below) to estimate the daily number of new infections, the plausible range of new infections, the cumulative number of infections, or the daily or cumulative infections as a percentage of the total population or per 1M persons.

Each of these covid19.Explorer applications uses a model—but it is one whose parameters are set by the user, rather than estimated from the data. In particular, users of the covid19.Explorer package or corresponding web interface will need to specify: (1) a value or set of values for the infection fatality ratio, IFR (Roques et al., 2020), of SARS-CoV-2 infection through time; and (2) an average lag time from infection to death. Each of these model parameters have been assigned default values that are fairly reasonable, as detailed in the sections below; however, users are nonetheless strongly encouraged to apply multiple values and examine the sensitivity of their results.

The second type of function (exemplified by the web application tabs Deaths by age, Excess mortality by age, and By state) do not employ an explicit model and exist primarily to permit the user to interact directly with CDC COVID-19 death and 2020 excess mortality data, to understand the implications of these data, and to generate interesting or useful data visualizations. The names and corresponding web application tabs (if applicable) of all functions in covid19.Explorer are given in alphabetical order in Table 1, above.

Table 1 A summary of the functions and corresponding web applications that currently make up the covid19.Explorer R package.

Function name	Application tab	Description	
age.deaths	Excess mortality by age	Graph weekly or cumulative excess mortality by age and jurisdiction.	
compare.infections	State comparison	Compare daily or cumulative deaths and estimated daily or cumulative infections between states and U.S. jurisdictions.	
covid.deaths	Deaths by age	Plot weekly or cumulative confirmed COVID-19 deaths by age group and compared to all deaths.	
iceberg.plot	Iceberg plot	Graph observed daily confirmed SARS-CoV-2 cases (above the ‘waterline’ of the graph) and estimated unobserved infections (below it).	
infection.estimator	Infection estimator	Estimate daily or cumulative SARS-CoV-2 infections based on observed deaths and confirmed cases.	
infection.range.estimator	Plausible range	Estimate the plausible range of daily or cumulative infections based on an interval of IFR values at each time point.	
infections.by.state	SARS-CoV-2 infections	Visualize geographic distribution of new or cumulative SARS-CoV-2 infections through time.	
state.deaths	By state	Graph weekly or cumulative excess deaths by U.S. state.	
updateData	Not applicable	Update the data used by covid19.Explorer from the web.	

Estimating infections

Since the beginning of this pandemic, it has been widely understood that confirmed COVID-19 cases underestimate the true number of infections, sometimes vastly (Al-Sadeq & Nasrallah, 2020; Wu et al., 2020). This underestimation has multiple causes. One important factor is that there has been limited testing capacity throughout much of the SARS-CoV-2 pandemic in the United States, but particularly when the pandemic was in its earliest days (Rosenberg et al., 2020). A second significant factor affecting the disconnect between observed cases and true infections are the facts that in the United States SARS-CoV-2 testing is voluntary, population surveillance testing has been relatively scarce, and many cases of SARS-CoV-2 infection present asymptotically or with mild symptoms (Oran & Topol, 2020). As such, I consider confirmed COVID-19 deaths to be a much more reliable indicator of disease burden than confirmed cases. Deaths, however, are a lagging indicator of infections.

The key parameter that relates daily COVID-19 deaths to the number of infections is the infection fatality ratio (also called the infection fatality rate or IFR). IFR, normally expressed as a percent, is defined as the fraction of deaths among all infected individuals, taking into account both observed infections (‘cases’) and asymptomatic or unobserved infections (O’Driscoll et al., 2020). An IFR value of 1.5%, for example, would mean that, on average, for every 1,000 infections in a specified population, there would be 15 deaths.

I modeled the number of new SARS-CoV-2 infections on the ith day by taking the number of observed COVID-19 deaths on day i + k (in which k is the average lag period between initial infection and death, where death is the outcome of infection), and then dividing this quantity by the IFR. In other words, given 50 COVID-19 deaths on day i + k, and an IFR of 0.5%, we would predict that 10,000 new SARS-CoV-2 infections had occurred on day i. Both k, the average lag time from infection to death (in cases of SARS-CoV-2 infections resulting in death), and the IFR are to be specified by the user.

A fairly reasonable lag time between infection and death might be approximately three weeks. For example, during a large outbreak in Melbourne, Australia the time difference between the peak recorded cases and peak confirmed COVID-19 deaths was around 17 days. Infected persons normally test negative for the first few days following exposure (Kucirka et al., 2020), so this more or less corresponds with a three week lag. Likewise, Wilson et al. (2020) report a median time from symptom onset to death of 13 days, and a meta-analysis by Dhouib et al. (2021) showed an incubation period of approximately 5–7 days, also corresponding to a lag time of approximately 18–20 days.

Likewise, IFR values ranging from about 0.2% to over 1.0% have been reported over the course of the pandemic. For instance, a study based on an early, super-spreader event in Germany estimated an IFR (corrected to the demographic distribution of the local population) of 0.36% (Streeck et al., 2020). Other researchers have reported higher estimated IFR (e.g., Rinaldi & Paradisi, 2020). In a large meta-analysis O’Driscoll et al. (2020) estimated IFR of SARS-CoV-2 infection across 45 different countries and obtained median estimates ranging from 0.24% to 1.49%, with higher IFRs typically reported for countries with older populations. In general, it is probably reasonable to suppose that IFR has fallen through time as treatment of severely ill patients has improved (Fan et al., 2020). Likewise, even within the U.S., IFR is unlikely to be precisely the same at a given date in different jurisdictions, due to differences in demographic structure between areas as well as other factors.

I suspect that it is within reason for users of covid19.Explorer to specify an IFR that is no greater than about 1.5% and that declines gradually from the start of the pandemic towards the present, with a current IFR that is perhaps around 0.3–0.5% (O’Driscoll et al., 2020; Blackburn et al., 2021). Nonetheless, covid19.Explorer permits the user to specify a time-varying IFR by fixing the IFR at each quarter (on the website), or at any arbitrary time interval (using the R package directly), and then interpolating daily IFR between each period using local regression smoothing (LOESS; Cleveland, 1979). As such, it is also possible to build a model for IFR through time that both falls and rises, perhaps as stresses on local healthcare resources increase or decrease through time with rising and falling COVID-19 case numbers.

Reporting can vary through time including regularly over the course of the week. (For instance, fewer COVID-19 deaths tend to be reported on the weekends compared to Monday through Friday; e.g., Fig. 1A) To take these reporting artifacts into account, I used both moving averages and local regression (LOESS) smoothing. Both the window for the moving average and the LOESS smoothing parameter are controlled by the user.

Figure 1 (A) Observed U.S. daily COVID-19 deaths (red bars) and user-specified infection fatality rate (IFR) function (blue line) through time. Note that this panel of the figure has two vertical axes. The axis on the left shows IFR in %, corresponding to the user-specified IFR model indicated by the blue curved line. The axis on the right shows the number of new daily COVID-19 deaths, corresponding to the vertical red bars. (B) The ratio of daily confirmed SARS-CoV-2 infections over estimated infections (grey points) and a fitted sigmoid function of the implied case detection rate (CDR) through time. This fitted curve is used to extrapolate the true number of daily new SARS-CoV-2 infections from reported cases in the most recent reporting days.

The approach of using only confirmed COVID-19 deaths—though robust—does not permit us to estimate the true number of infections between k days ago and the present. To do this, I assumed a sigmoidal relationship (by default) between time and the ratio of daily confirmed cases over the estimated true number of infections—a quantity called the case detection rate or CDR (Fig. 1B). Since the number of confirmed cases cannot exceed the true number of new infections, logic dictates that the CDR should have a value that falls between 0 and 1.

I decided on a sigmoidal relationship between the case detection rate and time because it seemed reasonable to presume the ratio was very low early in the pandemic when confirming a new infection was limited primarily by testing capacity, but that CDR has probably risen (in many localities) to a more or less consistent value as testing capacity increased. Since getting tested is voluntary, and since many infections of SARS-CoV-2 are asymptomatic or only mildly symptomatic, this ratio seems unlikely to rise to very near 1.0 in the U.S. regardless of the availability of testing. Fig. 1, created using covid19.Explorer, shows daily confirmed cases/daily estimated infections (under our model) for all U.S. data over the entire course of the pandemic to date (Fig. 1B), given observed daily deaths (red bars) and assumed IFR evolution through time (blue curved line; Fig. 1A). Our plot seems to indicate a CDR of about 0.42 at the present; however, the reader should keep in mind that in practice this value is estimated separately for each jurisdiction that is being analyzed, and as such might be lower in some states and higher in others, even for a constant IFR value or function.

In the event that a sigmoid function cannot be fit to the implied daily CDR for a given state or jurisdiction, the software automatically substitutes the mean CDR from the last 30 days of data. Since I only used the CDR to estimate daily infections for the most recent time period of our data (see below), and since CDR tended to increase asymptotically towards a more or less constant value in most jurisdictions (e.g., Fig. 1), this seemed fairly reasonable. When using the covid19.Explorer in R (rather than through the web interface), this option can also be selected explicitly by the user. An important point to make in this context is that I intend the sigmoidal functional form to be a heuristic (rather than literal) means of capturing the approximate relationship between CDR and time since the start of the pandemic—and thus estimate the CDR for the most recently reported cases. If users are unsatisfied with the fit of the sigmoid curve to CDR, they are encouraged to substitute the mean implied CDR from the last 30 days of data. The reason I chose the sigmoid fit to begin with was primarily to avoid distortions driven by so-called ‘data dumps,’ in which a state or jurisdiction releases a large number of previously misclassified or unreported cases or deaths on a single day. In practice, using the mean implied CDR from the past 30 days or the fitted value of CDR from a sigmoid fit will not make much of a difference in the majority of jurisdictions represented in our data.

After fitting this sigmoidal curve to our observed and estimated cases through now—k days (or calculating the mean implied CDR from the most recent 30 days), we then must turn to the last period. To obtain estimated infections for these days, we merely divide our observed cases from the last k days of data by the fitted CDR values of our curve. Figure 2 shows the result of this analysis applied to data for the U.S. state of Massachusetts.

Figure 2 (A) Observed daily COVID-19 deaths and an assumed model of IFR in which the infection fatality ratio is initially high (~1.5%), but then declines and stabilizes at around 0.6% through the present day. As in Fig. 1A has two vertical axes. The axis on the left shows IFR in %, corresponding to the user-specified IFR model indicated by the blue curved line. The axis on the right shows the number of new daily COVID-19 deaths, corresponding to the vertical red bars of the plot. (B) Estimated daily infections (green), cases (blue), and deaths (red).

In addition to computing the raw number of daily infections, this method can also be used to estimate infections as a percentage of the total population. To make this calculation, I obtained state populations through time from the U.S. Census Bureau. Data was only given through 2019 at the time of writing, so to estimate state-level 2020 population sizes, I used a total mid-year 2020 U.S. population estimate of (331,002,651) to ‘correct’ each 2019 state population size to a 2020 level.

Finally, CDC mortality data splits New York City (NYC) from the rest of New York state. Since this contrast is interesting (e.g., Gonzalez-Reiche et al., 2020), I maintained the separation—and used a mid-2019 population estimate of (8,336,817) for NYC, then simply assumed that the population of NYC has changed between 2015 and 2020 in proportion to the rest of the state. Since they have a part: whole relationship, this seemed pretty reasonable. In fact, according to the U.S. Census Bureau from 2010 to 2019 the fraction of New York State residents living in New York City is estimated to have grown by around 0.1% per year, from 41.8% in 2010 to about 42.8% in 2019. If this trend continued through 2020, then I may have underestimated the population of New York City by about 0.2%. Since this is only relevant when considering per capita SARS-CoV-2 infections and COVID-19 deaths, I suspect it is a relatively minor source of error compared to other simplifying assumptions of this software.

Assumptions about estimating infections

This model is very simple. In using it, we start by merely imagining that if we knew the true number of infections and the IFR for our population of interest on day i, then we could predict the number of deaths on day i + k, in which k is the lag-time from infection to death (for SARS-CoV-2 infections leading to death). Having observed the deaths, and supposing a particular value of IFR for day i, we can likewise work backwards and reconstruct the most plausible number of infections on that day.

Although the model does not pre-suppose a specific value or function for IFR, it does require that one be specified by the user. As such, it is probably worth mentioning the effect of setting an IFR value that is either too high or too low compared to the (invariably unknown) true IFR for the population of interest. An IFR that is too high (overall or at a specific time during the pandemic) will have the general effect of causing us to systematically underestimate the number of infections that have occurred. This makes sense because if we imagine observing 50 COVID-19 deaths, an IFR of 0.5% would imply that these deaths correspond to a total of 10,000 SARS-CoV-2 infections. By contrast, a higher IFR of, say, 1.0% would instead imply that only 5,000 infections had occurred. Assuming an IFR value that is too low will (obviously) have exactly the opposite effect and thus cause us to overestimate the number of infections that have occurred. The default values for IFR through time specified in the web portal (https://covid19-explorer.org) are 0.85% on February 1, 2020 and then decline every 3 months: 0.65%, 0.55%, 0.5%, and 0.5% on January 31, 2021, with intermediate values interpolated using LOESS smoothing.

The purpose of the software and web resource is to allow the user to explore alternative (reasonable) scenarios for IFR through time and examine their effects on estimated daily or cumulative SARS-CoV-2 infections in different jurisdictions; however, the default values are not arbitrary. First, they are largely consistent with population-wise IFR estimates from seroprevalence research (e.g., O’Driscoll et al., 2020). Second, they yield estimated daily infections that are qualitatively if not quantitatively similar to those obtained by several other leading models of the SARS-CoV-2 pandemic in the United States (e.g., Gu, 2020; Reiner, Barber & Murray, 2020).

I also assume a homogeneous value of k at any particular time. In fact, literature sources report lag-times between two and eight weeks (e.g., Yang et al., 2020). Nonetheless, I suspect that inferences by this method should not be badly off—so long as the true IFR does not swing about wildly from day to day, and so long as the number of deaths is not extremely few for any reporting period. I likewise assume a constant lag-period, k, through time. This assumption is perhaps a bit more dubious as it seems quite reasonable to suppose that, for a specific state or jurisdiction, as IFR falls k might also increase. If k increased as a function of time, this would mean that recent peaks in daily new infections would be systematically biased forward in time (that is, they occurred earlier than it seems) compared to peaks that occurred early in the pandemic. (The converse would also be true if k decreased rather than increasing through time.) This is a complexity that I explicitly chose to ignore in the model.

I assume that a more or less consistent fraction of COVID-19 deaths are reported as such—that is, that COVID-19 is neither systematically under- or overreported as the cause of death at any point during the course of the pandemic. A violation of this assumption is not quite as grave as it might seem, however, because it can simply be ‘baked in’ to our model for IFR. For instance, if we think that COVID-19 deaths were under-reported near the start of the pandemic (e.g., Weinberger et al., 2020), perhaps due to limited testing capacity, this can be accommodated into our model for daily infections simply by specifying a slightly lower IFR value for SARS-CoV-2 infection at that time (keeping in mind, of course, that the true IFR has generally decreased through time; e.g., Levin et al., 2020).

In estimating the number of daily infections from k days ago to the present, we assume that the relationship between time (since the first infections) and the ratio of confirmed and estimated infections (i.e., the case detection rate, CDR) is sigmoidal in shape (Fig. 1B). This is a testable assumption that seems to hold fairly well across the entire U.S. (Fig. 1B) and for some jurisdictions, but less well for others. It is equally plausible to suspect that CDR could shift not only as a function of time, but also as demands on testing capacity rise and fall with case numbers, or as different populations become infected. This should be the subject of additional study, but my suspicion is that this would not be likely to have a large effect on our model compared to other simplifications. Additionally (as mentioned above), when using covid19.Explorer from within R it is straightforward to substitute the mean implied CDR from the last 30 days for the fitted values of CDR from the sigmoidal fit.

One slightly problematic possibility is that the true CDR in the most recent k days is much lower or higher than estimated CDR. This could happen if, for example, in jurisdictions with low surveillance testing, changes in the demographic distribution of new SARS-CoV-2 infections (due to, for instance, age-prioritized vaccination) mean that relatively few infections present symptomatically and get tested. This would have the effect of causing CDR to be overestimated and would result in a concomitant underestimation of daily new SARS-CoV-2 infections towards the right side of the graph. The opposite effect is expected if surveillance testing was to be increased (for instance, in a jurisdiction with high numbers of in-person college or university students simultaneously returning to campus), thus increasing true CDR relative to its estimated value in the most recent period compared to time periods prior to k days before the present.

Finally we assume no or limited reporting delay. This is obviously incorrect. There are two main sources of reporting delay: the delay between when an individual is infected and when they go on to test positive for SARS-CoV-2; and the delay between when an infected patient dies and their death is reported to the CDC. Given this delay in reporting, a more precise interpretation of the estimated number of daily infections, is a (rough) estimate of the number of new individuals who would be reported as testing positive for SARS-CoV-2 on any given day under a hypothetical scenario of universal daily testing.

Showing observed and estimated unobserved infections using an ‘iceberg plot’

As noted above, it has long been well-understood that the number of daily confirmed COVID-19 cases is an underestimate of the true number of daily SARS-CoV-2 infections, sometimes by a very wide margin (Wu et al., 2020). To visualize this phenomenon, I devised an iceberg plot in which we simultaneously graph the number of observed infections (above the ‘waterline’ of the graph) and the estimated number of unobserved SARS-CoV-2 infections (below it). Figure 3 gives this analysis for New York state, in which I assumed the same IFR model through time as was used to generate Figs. 1 and 2.

Figure 3 Iceberg plot showing the confirmed daily new infections (above the ‘waterline’ of the plot) and estimated unobserved infections (below it) for New York state.

Mapping the distribution of infections across states

A hallmark feature of the U.S. COVID-19 pandemic has been the shifting geographic distribution of infections through time among states. To capture this dynamic, I devised a plotting method for covid19.Explorer in which I overlay the daily or cumulative SARS-CoV-2 infections under our model (outlined above), separated by state.

For this visualization, I selected a geographic color palette such that RGB color values were made to vary as a function of latitude, longitude, and (arbitrarily) geographic distance from Florida. This is intended to have the effect of making the regional geographic progression of infection more apparent in the graph. The result can be seen in Figs. 4 and 5.

Figure 4 Daily estimated infections separated by state.

The color palette is designed to capture the geographic distribution of new infections through time, rather than the severity of the pandemic in each state.

Figure 5 The covid19.Explorer web interface (https://covid19-explorer.org) showing estimated cumulative SARS-CoV-2 infections among states under the same IFR model as Figs. 1–4.

This plotting method shares all the assumptions of our infection estimator, above, but adds the additional assumption that our model of IFR is the same for all states. This assumption is quite dubious, in fact, as IFR could be expected to rise in locations were hospital resources are overtaxed by high disease burden; and, conversely, fall in hospitals where staff have more experience in treating COVID-19 patients.

On an individual level, IFR is also very strongly influenced by age (e.g., O’Driscoll et al., 2020), as well as by other risk factors such as obesity (e.g., Kompaniyets et al., 2021) and socioeconomics (e.g., Lone et al., 2021). As such, even if IFR falls through time in different jurisdictions in a similar way, one would nonetheless expect to observe higher IFR in states with higher median age, higher obesity, or higher poverty rates, compared to younger, less obese, and higher median income states. Although I do not doubt that these nuances are important in making specific, quantitative statements about the particular number of infections in each state, I nonetheless believe that my method is effective at visually capturing the overall geographic dynamics of the COVID-19 pandemic in the United States.

One point that may be worth noting about this lattermost assumption is that use of a constant IFR model across all U.S. states does not, in and of itself, have the effect of distorting the total number of estimated new infections on each day. To see this, let’s start by imagining (for instance) 3,000 infections in jurisdiction A on day 1 and 30 resultant deaths k days later (IFR of 1.0%). Meanwhile, perhaps, 2,000 infections have occurred in jurisdiction B on day 1, but only 10 deaths k days later (IFR of 0.5%). Using the global IFR of (30 + 10)/(3,000 + 2,000) × 100% = 0.8% gives us the same estimate of the total number of new infections on day 1 (40/0.008 = 5,000) whether it is applied to each jurisdiction separately, or to the total number of deaths taken all together. What is affected, however, are the proportions of new infections attributed to each jurisdiction. In the constant IFR model the number of infections attributed to jurisdiction A (30/0.008 = 3,750) would be too few; while the number of new infections attributed to jurisdiction B (10/0.008 = 1,250) is too many. Thus the distribution of daily new infections among sites, but not their grand total across jurisdictions, can be affected by an assumption that the IFR of SARS-CoV-2 (and the way that it changes through time) is the same across all of the jurisdictions in our dataset.

Visualizing COVID-19 mortality data

In addition to modeling the number of infections through time, the covid19.Explorer R package and website also allows users to visualize the distribution of COVID-19 deaths by age and sex, as well as mortality in excess of normal during 2020 compared to other recent years (2015–2019).

Excess mortality (also called mortality displacement; e.g., Huynen et al., 2001) is defined as the number of deaths (for any period) in excess of the ‘normal’ number of deaths for the same period. To compute the raw death counts for each jurisdiction, I tabulated the 2015–2018 counts with the 2019–2020 provisional counts. To correct observed deaths in prior years to 2020 levels, I simply multiplied the past-year death tally by the ratio the jurisdiction population in 2020 compared to the population in the past year. Finally, to compute excess deaths for any jurisdiction, I then took the death counts (or corrected death counts) for 2020, and subtracted the mean of years 2015 through 2019. This treats 2015 through 2019 as ‘normal’ years, and 2020 as unusual.

One factor that I did not account for in this lattermost calculation is movement of people between jurisdictions. In fact, some studies indicate that the COVID-19 pandemic has disrupted normal immigration patterns of humans (e.g., Smith & Wesselbaum, 2020). Areas harder-hit by SARS-CoV-2 may have experienced a net loss of residents (even apart from direct mortality due to COVID-19) due to emigration of people from the affected jurisdiction, or reduced immigration to the area (Smith & Wesselbaum, 2020). Fortunately, the covid19.Explorer R package and web application will be easy to update when final census and estimated population sizes for the states and jurisdictions of my dataset are published for 2020 and 2021.

The covid19.Explorer web interface

Though the covid19.Explorer package can be used within an interactive R session, it has also been interfaced to the web by way of the web application that I developed in Rstudio (RStudio Team, 2020) using the shiny web development system (Chang et al., 2021). The covid19.Explorer web application is hosted at the website https://covid19-explorer.org.

Figure 5 shows a screenshot of this web application, illustrating an analysis of the estimated cumulative number of SARS-CoV-2 infections through time across U.S. states. In this web application, the user must specify the value of IFR at the beginning of each 3 month period, and that at the end of the year, beginning on February 1, 2020, and ending on January 31, 2021. Values on these intervals are interpolated using LOESS smoothing.

Although the default values for the IFR of SARS-CoV-2 and the average lag time from infection to death on the web interface are somewhat arbitrary (and are meant to be adjusted by the user), they both fall on the range of most estimated values for these parameters from other research (e.g., O’Driscoll et al., 2020; Wilson et al., 2020), and result in estimated daily new SARS-CoV-2 infections that are qualitatively and/or quantitatively similar to other leading resources (e.g., Gu, 2020; Reiner, Barber & Murray, 2020).

Results

The purpose of this article is to describe a software tool, which I have largely done in the preceding section. Here, I will attempt to highlight some results and insights that can be obtained by users via interaction with the covid19.Explorer R package or web application.

Herd immunity and the cumulative proportion of the population infected

The question of cumulative percent infected is relevant to the concept of ‘herd immunity’ (Randolph & Barreiro, 2020). The herd immunity threshold (HIT), whether reached via natural infection or vaccination, is typically defined as the proportion of the population that must be immune in order to cause the basic reproductive number of the virus at time t (Rt) to fall below 1.0, absent mitigations (Anderson & May, 1985). When Rt has fallen below 1.0, daily new infections should progressively decline.

The HIT is normally estimated by taking the reproductive number when 100% of the population is susceptible (i.e., when a new disease emerges, R0), and computing 1 − 1/R0. For SARS-CoV-2 various values of R0 have been represented in the literature, from as low as around R0 = 2.4 (e.g., D’Arienzo & Coniglio, 2020), to as high as about R0 = 5.8 (e.g., Ke et al., 2020). A value of R0 equal to 3.0, for example, would imply that herd immunity should be reached after 1 − 1/3 or around 67% of the population has acquired immunity through natural infection or vaccination (not accounting for waning acquired immunity from natural infection, which some studies have indicated for SARS-CoV-2; e.g., Long et al., 2020).

The covid19.Explorer R package and web application can be used to evaluate the proportion of individuals in the total population that have been potentially infected with SARS-CoV-2, given our model for COVID-19 IFR through time. Figure 6 shows cumulative estimated SARS-CoV-2 infections as a fraction of the total population for the U.S. state of Texas, using the same IFR model as in Figs. 1–4. Though the plot suggests that perhaps around 25–30% of the population in Texas has already been infected, users should keep in mind that this result is entirely dependent on how we decided to specify our model of IFR through time. Likewise, though this fraction is considerable, it is still well below the level of infection (e.g., 67%) required to achieve herd immunity given the majority of published estimates for R0 of SARS-CoV-2. It may be worth noting that some authors have pointed out that the herd immunity threshold from a natural epidemic could be considerably lower than the 1 − 1/R0 level expected for random vaccination (e.g., Britton, Ball & Trapman, 2020; Gomes et al., 2020). This is an intriguing possibility, and one that could be qualitatively examined with some of the tools of the covid19.Explorer package.

Figure 6 (A) Observed daily COVID-19 deaths and an assumed model of IFR. As in Figs. 1 and 2, panel A has two vertical axes. The axis on the left shows IFR in %, corresponding to the user-specified IFR model indicated by the blue curved line. The axis on the right shows the number of new daily COVID-19 deaths, corresponding to the vertical red bars of the plot. (B) Estimated cumulative SARS-CoV-2 infections (green), cases (blue), and deaths (red), as a percentage of the total population of the state.

Computing a plausible range of infection numbers

A relatively simple extension of the infection estimation method, described above, is to admit uncertainty about the specific value of the infection fatality rate at any particular time during the pandemic, and then measure the sensitivity of our prediction to a wide range of different values for IFR.

The question of the IFR for COVID-19 has been the subject of considerable controversy and confusion (e.g., Vermund & Pitzer, 2020). This model can be design to accommodate an assumption of broad uncertainty in IFR early during the pandemic, with both decreasing IFR, as well as decreasing uncertainty in IFR, towards the present. This is illustrated for data from the U.S. state of Louisiana in Fig. 7.

Figure 7 (A) Confirmed COVID-19 deaths and a plausible range of scenarios for the evolution of SARS-CoV-2 infection fatality rate (IFR) through time. As in Figs. 1, 2, and 6 panel A has two vertical axes. The axis on the left shows IFR in %, corresponding to the user-specified IFR model (in this case, given as a plausible range of IFR values for each time period) indicated by the blue curved line and shaded area. The axis on the right shows the number of new daily COVID-19 deaths, corresponding to the vertical red bars of the plot. (B) A corresponding plausible range of daily new infections, under our model, for the U.S. state of Louisiana.

It should be noted that although the shaded region around the mean number of daily or cumulative infections in Fig. 7 looks like a confidence band, it would only be valid to consider it as such if our high and low values of the IFR through time represented a confidence interval around the true infection fatality rate (and, even then, this confidence band would only take into account one source of uncertainty about the real daily number of infections—the IFR). As an increasing number of studies are able to provide us with better and better estimates of the IFR of SARS-CoV-2 throughout this pandemic (e.g., O’Driscoll et al., 2020) it may be possible to parameterize this model in a way that genuinely accounts for changing uncertainty in the value of IFR through time in the U.S. pandemic. For the time being, however, I recommend employing the method as a heuristic approach to obtaining a credible range of daily new or cumulative SARS-CoV-2 infections under an explicit model for the United States or any particular U.S. jurisdiction.

Comparing daily and cumulative infections between states

Another straightforward extension of our above-described model involves directly comparing daily or cumulative infections between states. This, likewise, could be a useful activity because many readers have undoubtedly observed how common it has become for popular press sources to attribute different infection dynamics in different states to one public health intervention or another. This attribution may be valid in many instances, but is often confounded by varying infection dynamics through time in the different states being compared. In general, evaluation of non-pharmaceutical interventions on the spread of SARS-CoV-2 (e.g., Bennett, 2021; Liu et al., 2021b) has been both very difficult and problematic. In Fig. 8, I compare the daily confirmed deaths and estimated infections between the U.S. states of California and Florida.

Figure 8 Daily confirmed COVID-19 deaths (A) or estimated SARS-CoV-2 infections (B) in the U.S. states of California vs. Florida.

This plotting method obviously shares all of the assumptions of our infection estimator, and (just like our method for visualizing the geographic dynamics of the pandemic across all U.S. states) requires that we use the same IFR model for each state. Since the daily and cumulative number of infections scales with population size, valid state-to-state comparisons really only make sense if done on a per-capita basis (e.g., infections or deaths/1M population), just as shown here in Fig. 8.

COVID-19 mortality and age

Lastly, in addition to modeling the number of SARS-CoV-2 infections through time, the covid19.Explorer package can be used to analyze and graph COVID-19 deaths by age and sex, as well as excess mortality by age and jurisdiction.

This functionality, too, can sometimes lead to valuable insights. For instance, it was widely predicted by media and public health experts that school and college reopening in the fall was likely to increased SARS-CoV-2 infections and increased COVID-19 deaths among U.S. children and young people, as well as increased SARS-CoV-2 transmission in the community (e.g., Bansal, Carlson & Kraemer, 2020). In my opinion, the minimum standard of evidence required to establish that reopening of colleges and universities for the fall semester of 2020 had led to increased community transmission overall (remembering the adolescents and young adults live in communities, regardless of whether they are on campus or at home) would be increased SARS-CoV-2 infections of college-aged youth, as a proportion of all infections, during the fall than in spring or summer.

In fact, and keeping in mind that, prior to widespread vaccination, COVID-19 deaths are always a better (though lagging) indicator of SARS-CoV-2 infections than observed cases (Gu, 2020; Reiner, Barber & Murray, 2020), CDC mortality data show precisely the opposite pattern. Figure 9 gives the weekly COVID-19 deaths over all ages (in panel a) and for 15–24 year olds (in panel b). We see that although the highest peaks of weekly COVID-19 deaths in the general population occurred in the spring of 2020 and the fall/winter of 2020/21, peak deaths among 15–24 are similar between summer and fall, and much higher as a proportion of all COVID-19 deaths during the summer—precisely when schools and colleges were out of session for all students.

Figure 9 Weekly confirmed COVID-19 deaths for (A) all ages; and (B) individuals aged 15–24 years old in the U.S.

The pattern is quite unambiguous: as a fraction of all COVID-19 deaths, weekly COVID-19 deaths among 15–24 year olds were nearly twice as high during the summer compared to Fall/Winter SARS-CoV-2 waves (Fig. 9). This pattern, however, has multiple interpretations. It is least consistent with increased SARS-CoV-2 infections among youth and young adults following school and university reopening; however, it is also consistent with disproportionately better clinical outcomes among young adults compared to other age groups in the Fall/Winter of 2020/21 compared to prior waves of SARS-CoV-2 infection. Since I know of no study indicating this, I find the latter explanation less plausible. On the other hand, many colleges and universities implemented aggressive COVID-19 interventions, such as bulk rapid-testing, mask-wearing, and widepsread contact-tracing (e.g., Mukherjee et al., 2021). This might in turn have meant that adolescents and college-aged adults were more (rather than less) likely to become infected with SARS-CoV-2 when schools were in summer recess then when they returned to campus in the fall.

Discussion

The SARS-CoV-2 global pandemic of 2020 and 2021 has upended economies and civil society worldwide. With widespread vaccination campaigns underway in many countries, and particularly in the United States, the COVID-19 pandemic may finally be in its waning days (even if SARS-CoV-2 ultimate becomes endemic and never entirely goes away, e.g., Shaman & Galanti, 2020). Nonetheless, understanding the temporal and geographical dynamics of SARS-CoV-2 infections and COVID-19 deaths remains a critically important endeavor. The COVID-19 pandemic is neither the first, nor will it be the last, global respiratory virus pandemic (Saunders-Hastings & Krewski, 2016; Piret & Boivin, 2021). Lessons learned from this pandemic will be of substantial and lasting consequence in managing or failing to manage future public health emergencies.

In this article, I present an accessible tool—the covid19.Explorer R package and corresponding web application—that is designed to be used to model U.S. SARS-CoV-2 infections through time, to understand the differences in epidemic dynamics between states and jurisdictions, to visualize the geographic progress of infection among U.S. states, to graph confirmed COVID-19 deaths by age and sex, and to compute and visualize excess mortality by age and jurisdiction.

Given the impact the SARS-CoV-2 pandemic has had on almost all of our daily lives over the past year, most readers of this article will know (or will be unsurprised to learn) that many other software tools and web-based applications have been developed to help visualize or better understand the temporal or geographic dynamics of COVID-19 in the United States. I nonetheless believe, however, that covid19.Explorer application, which has now been online (in one form or another) for nearly 7 months, contains a number of different functionalities and graphics not readily available in other competing tools.

Firstly, no other software or web application, to my knowledge, lets the user build a custom model for the evolution of infection fatality rate through time. This facility, offered by covid19.Explorer, allows the scientists and members of the general public that interact with the software to design their own parameter function (be it based on specific hypothesis about IFR through time, or external information—e.g., from seroprevalence studies—about the value of IFR for SARS-CoV-2 at a specific time and place) that will then be used to estimate infections under the model. Likewise, the tool allows covid19.Explorer users to progressively adjust the parameter values and other assumptions of this model and see how their results change in turn. Secondly, multiple visualization methods of the covid19.Explorer R package and webpage, such as Fig. 4 showing the geographic distribution of estimated SARS-CoV-2 infections through time or the ‘iceberg’ plot of Fig. 3, are simply not represented in other software packages. Lastly, the covid19.Explorer package is completely transparent and open source. It pulls its data directly from public, government repositories. All model assumptions (even those not explicitly described in this paper) are readily identified from the software source code of the package functions.

Even if the SARS-CoV-2 pandemic eventually becomes a distant memory, I hope that this tool (which I plan to make available indefinitely) will continue to be of use to scientists and educated members of the public interested in the learning from the successes and failures of policy during the 2020/21 pandemic—perhaps to ensure that there are more of the former and fewer of the latter in our next global infectious disease pandemic.

This work is inspired by the brilliant, independent COVID-19 research of Y. Gu; by the calm and thoughtful public health insights of J. Allen, F. Balloux, S. Baral, M. Gandhi, A. Munro, and others like them; by my wife, E. Lu, who has been forced (due to stay at home orders and other restrictions) to suffer much more of my research struggles than she would under normal circumstances. The article was improved greatly over earlier versions due to numerous helpful comments from N. Dimonaco, A. Kala, and one anonymous reviewer.

Additional Information and Declarations

Competing Interests

Author Contributions

Data Availability

The authors declare that they have no competing interests.

Liam J. Revell conceived and designed the experiments, performed the experiments, analyzed the data, prepared figures and/or tables, authored or reviewed drafts of the paper, and approved the final draft.

The following information was supplied regarding data availability:

Code and data are available at https://github.com/liamrevell/covid19.Explorer/.

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
