# Peer review of "covid19.Explorer: a web application and R package to explore United States COVID-19 data"

_PeerJ, doi:10.7717/peerj.11489_

## Round 0.1 · original submission · Major Revisions

Dear author, my congratulations on an interesting manuscript, please follow the reviewers' observations to improve quality.

·

Basic reporting

Abstract:
The first sentence could be rewritten. ‘Beginning at the end’ reads a little oddly for the intro of the abstract. Maybe something like “Appearing at the end of 2019…”
Remove ‘I Hope’ from last sentence. Make the abstract confident of the work presented.

Introduction:
Paragraph 2: R is also a programming language and not just an environment. - Insignificant point
Introduction is too short.
Can you please provide information on other examples of similar systems and how this improves upon those such as: https://coronavirus.jhu.edu/map.html / https://covid19.who.int/ / https://www.worldometers.info/coronavirus
Can you expand the aim of this project/paper?

Experimental design

Methods:
1.3: You are repeating some points from above. I understand why but the use of the same citations makes it too repetitive on the same page. Can you remove or shorten the first mention of case underreporting?

"As such, I consider confirmed COVID-19 deaths to be a much more reliable indicator of disease burden than 94 confirmed cases” : I imagine there are others which agree here so could you cite something such as https://www.nature.com/articles/s41586-020-2918-0

I understand these are estimations or examples but using an estimated ‘IFR’ ratio in the middle of another estimation function could be clarified further. Maybe detail IFR first?

“covid19.Explorer permits the user to specify a time-varying IFR by fixing the IFR at each quarter” - This could be linked to outbreaks and stress on local health services. IFR are likely to be demographically dependant. For example, some states have older and poorer populations but if a single IFR is used across the US, can this have influence?

Line 129: “Since the true number of new infections cannot exceed the number of confirmed cases, logic dictates that this ratio must have a value between 0 and 1.” I do not understand. Please clarify.

Line: 154: “.... simply assumed that the population of NYC 155 has changed between 2015 and 2020 in proportion to the rest of the state.” I understand the reasoning but NYC is a very different place with very different attractions as compared to NY-State. Is there anything to back this up?

Section 1.3/1.4:
Did you look at other types of regression now that we have so much data over the past year?

The paper states that small or arbitrary changes in IFR have major unfluence on the model output but the system entirely hands over the responsibility of selecting an appropriate IFR without much guidance or example (Specifically for the US). I think IFR selection could do with some defaults computed from some previous studies, some already reported in the paper. The website is presented to a user with defaults but they are not described.


Line 179 - There have been a number of studies into the under or over reporting of COVID-19 related deaths - There are many points in the methods which require citation (example https://jamanetwork.com/journals/jamainternalmedicine/fullarticle/2767980)
Listed are a number of assumptions are why they are mostly incorrect or incomplete. Can you comment more on the impact of these assumptions?

Section 1.6:
Noted again are how some assumptions are likely not adequate but not backed up or taken further. Please either rewrite or state evidence for their continued use after discreditation.

Section 1.7:
If you take into account the jurisdiction population for computing excess deaths, you may need to take into account state-wise migration events throughout the pandemic. Cities and some regions have seen large levels of migration: https://academic.oup.com/jn/article/150/11/2855/5896932?login=true

Validity of the findings

Results:
Section 2.1:
Herd Immunity often does not take into consideration loss of antibodies and T-cells over time and therefore this 67% population figure is not the full picture: https://www.nature.com/articles/s41591-020-0965-6

Section 2.2:
Figure 7.a: Can you expand on why the IFR does not correlate well to the second and third death ‘spikes’?

Section 2.4:
Deaths of young adults 15-24 peaking while out of session in the summer have the influence of more outdoor activities and therefore more ‘risky’ situations. This same age group is seen to be little medically affected by SARS-2, therefore a better analysis would be of other age groups as the 15-24 age group would act as a vector of transmission to these older groups.

Discussion:
Line 317: Minor point - The pandemic is ongoing and likely to persist past 2021.

Line 320: Please elaborate why “understanding the temporal and geographical dynamics of SARS-CoV-2 320 infections and COVID-19 deaths remains a critically important endeavor”? (I agree but why?)

Line 335 “(based a hypothesis or external information)” - Do you mean “based on”?

Line 336: Assumptions are also parameters in this context?

Additional comments

Final Points:
Especially for a web-service, there should be an option for colour-blind friendly colours.

Please give some detail for the default parameters presented to the user on the website.

·

Basic reporting

The research article is well structured and written in easy to understand language for international readers.
The author has provided all the supplemental materials (including codes).
The figures are legible and of good quality for publishing.

One very minor edit to suggest is:
In some sections the author has addressed the steps as 'I have done this....' while in the other section he has addressed as 'We have done this....' - its better to be consistent.

Experimental design

The research, code and web product are very useful and definitely adds up a lot to the ongoing research in this field.

Author has laid the whole paper in a manner that is easy to understand and implement, with flexibility to customize the parameters and build your own insights.

Various graphical outputs of different conditions gives an outlook based on current data and future scenario - which is very useful from a research point as well as from decision making aspect.

Validity of the findings

The results, discussions and overall findings have a good basis to support with proper references (including recent ones).

The tool will be very useful to build new insights and initiate future advanced research.

Additional comments

Great work - looking forward to see your published articles.

Reviewer 3 ·

Basic reporting

1. What are the advantages of covid.Explorer over the existing R packages that also aimed to provide analysis and visualization of the COVID-19 data? What has been missing in those R packages that the author identified and filled in this covid.Explorer. I would suggest the author to provide sufficient literature review of the existing packages and explicitly identify the knowledge gap in the field in the introduction.

Experimental design

2. Line 129 “Since the true number of new infections cannot exceed the number of confirmed cases…” Does the author mean the opposite?

3. Could the author explain in Figure 1 a) and Figure 2 a), what are the models for the blue lines.

4. For the latest k days estimates of new infections, has the author considered, on top the estimated ratio of confirmed cases to new infections from the sigmoid shape, any other factors that may change that ratio? And how would that affect the accuracy of the estimates.

5. Could the author give more details on estimating the confidence interval of IFR?

Validity of the findings

6. Could the author give a detailed description of the U.S. data? What are the descriptive statistics of the characteristics of the data that has been used to illustrate the point?

Additional comments

In this manuscript, Revell developed a flexible R package and web portal to analyze and visualize the COVID 19 data of the United States. It has the ability to estimate the “true” cases on a certain date based on the number of deaths from a date that is k days before.

---

## Round 0.2 · Minor Revisions

Dear authors, I have reviewed and observed that you have substantially improved the manuscript and have included the comments of the reviewers, finally I ask you make minor revisions,

·

Basic reporting

no comment

Experimental design

no comment

Validity of the findings

no comment

Additional comments

The corrections and changes made in accordance with the first review have been much appreciated.
I am very happy with the current manuscript and have no further comments to make.

Reviewer 3 ·

Basic reporting

1. Line 141 no need to spell out the full name of IFR

2. Line 178 “Figure 1)” -> Figure 1 b).

3. The caption of figure 1 b) where the author described the CDR, “Ratio of daily
estimated infections over confirmed SARS-CoV-2 infections (grey points … (CDR)”, I think the author really means the opposite?

4. Line 187-189, “…shows daily estimated infections (under our model) / confirmed cases for all U.S. data…”, I guess the author would want to say “…confirmed cases / estimated infections (under our model)…”?

5. Line 194, “…a constant IFR value or function.” I don’t follow what is the relationship between CDR and IFR, could the author explain?

6. Figure 4 and 5, depending on how difficult it is, could the author make the color of states by the # of estimated infections, rather than by the spatial relationship to the state of Florida? It’s not intuitive to understand the color when it comes to the severity of the pandemic by states.

7. Line 376 add “at time t” for definition of R_t

Experimental design

1. I am still having trouble understanding figure 1 a). What does the author want to illustrate with this figure? E.g: is the author saying that the IFR is changing over time or changing over # of deaths? Is this trend over all the U.S. or specific to any region? How about population structure? Where does this user defined IFR come from? Has it been suggested from literature? Has the author estimated it from data? I would suggest the author to be more specific. Also the label of y axis and legend needs to be corrected accordingly.

2. Line 181-186, a suggestion that would make the argument stronger, that the author could provide a goodness-of-fit statistics on how good that sigmoid function fits the observed CDR over time. And it will be helpful to understand the sigmoid relationship if the author could give the details of the model, for example, the covariates and coefficients.

Validity of the findings

No comment

Additional comments

I think the author has responded to my comments well. The manuscript is in good shape, with sufficient background, assumption and description of the results. Only some minor revisions is needed. I am looking forward to the publication of this manuscript.

---

## Round 0.3 · accepted · Accept

Dear author in this round only reviewer 3 requested additional changes and I see that it has been much improved and my decision is Accept. My sincerest congratulations.